# Genome-Wide Identification of PYL/RCAR ABA Receptors and Functional Analysis of *LbPYL10* in Heat Tolerance in Goji (*Lycium barbarum*)

**DOI:** 10.3390/plants13060887

**Published:** 2024-03-20

**Authors:** Zeyu Li, Jiyao Liu, Yan Chen, Aihua Liang, Wei He, Xiaoya Qin, Ken Qin, Zixin Mu

**Affiliations:** 1College of Life Sciences, Northwest A&F University, Yangling 712100, China; lzy0107@nwafu.edu.cn (Z.L.); liujiyao7733@foxmail.com (J.L.); chwanyan@163.com (Y.C.); zombiessama@163.com (W.H.); 2College of Life Sciences & Technology, Tarim University, Alaer 843300, China; aiai_1025@163.com; 3State Key Laboratory Breeding Base for the Protection and Utilization of Biological Resources in Tarim Basin Co–Funded by Xinjiang Corps and the Ministry of Science and Technology, Alaer 843300, China; 4National Wolfberry Engineering Research Center, Ningxia Academy of Agriculture and Forestry Sciences, Yinchuan 750002, China; qinxiaoya@whu.edu.cn

**Keywords:** *Lycium barbarum*, ABA signaling, PYL/RCAR ABA receptors, genome–wide, heat stress

## Abstract

The characterization of the PYL/RCAR ABA receptors in a great deal of plant species has dramatically advanced the study of ABA functions involved in key physiological processes. However, the genes in this family are still unclear in *Lycium* (Goji) plants, one of the well–known economically, medicinally, and ecologically valuable fruit crops. In the present work, 12 homologs of *Arabidopsis* PYL/RCAR ABA receptors were first identified and characterized from *Lycium* (L.) *barbarum* (LbPYLs). The quantitative real–time PCR (qRT–PCR) analysis showed that these genes had clear tissue–specific expression patterns, and most of them were transcribed in the root with the largest amount. Among the three subfamilies, while the Group I and Group III members were down–regulated by extraneous ABA, the Group II members were up–regulated. At 42 °C, most transcripts showed a rapid and violent up–regulation response to higher temperature, especially members of Group II. One of the genes in the Group II members, *LbPYL10*, was further functionally validated by virus–induced gene silencing (VIGS) technology. *LbPYL10* positively regulates heat stress tolerance in *L. barbarum* by alleviating chlorophyll degradation, thus maintaining chlorophyll stability. Integrating the endogenous ABA level increase following heat stress, it may be concluded that LbPYL–mediated ABA signaling plays a vital role in the thermotolerance of *L. barbarum* plants. Our results highlight the strong potential of *LbPYL* genes in breeding genetically modified *L. barbarum* crops that acclimate to climate change.

## 1. Introduction

The phytohormone abscisic acid (ABA) plays a vital role in nearly all plant physiological processes under both normal conditions and stress conditions [1]. In comparison to its biosynthesis and catabolism, ABA signaling sense and signaling transduction may be more fundamental because there are more complicated regulation mechanisms that have evolved for the latter [2]. The most characteristic ABA receptor to date is a family of soluble proteins called PYR (pyrabactin resistant), PYL (PYR–like), or RCAR (regulatory components of ABA receptors), which form the beginning of a “core ABA signaling pathway” [3,4]. The pathway includes a PYL/RCAR ABA receptor, a negative regulator of clade A protein phosphatases of type 2C (PP2C), and a positive regulator of sucrose nonfermenting–1 (SNF1)–related protein kinase 2s (SnRK2s), which have been identified and characterized in a variety of plant species [5,6,7]. In this model, ABA binding stabilizes the receptor complex; as a result, the active site of the phosphatase is blocked and the associated protein kinase is released from PP2C inhibition. The activated protein kinase further activates downstream transcription factors and plasma membrane proteins to regulate gene expression and stomatal closure, respectively [8,9]. Genetic modification the coding genes of PYL/RCAR ABA receptors can improve plant water use efficiency in *Arabidopsis*, wheat, and maize, presenting a prospective target for engineering drought–tolerant crops under changing environmental conditions [10,11,12]. Moreover, chemical manipulation of ABA signaling by ABA agonists [13,14] and direct modification of PYL/RCAR ABA receptors by protein engineering [15] all have significant agricultural and ecological applications.

The ambient air temperature caused by global warming continues to rise, which makes heat stress (HS) even more serious [16,17,18]. When the temperature is higher than 38 °C, it will inhibit the growth of many crops, including rice, corn, wheat, etc., especially the pollen fertility of many crops will be reduced, and then the seed setting rate will decline, resulting in a significant reduction in crop yield. It will also cause grain filling not to be full, so that the grain quality is significantly reduced. HS severely inhibits photosynthesis and reduces carbohydrate synthesis by reducing the enzyme activity of the chloroplast protein complex. At the same time, the respiration of the plant will become strong, decompose, and consume a lot of organic nutrients, resulting in the “chlorina” and premature aging of the leaves [16,17,18]. The heat stress response (HSR) of plants is initiated by increased plasma membrane mobility and the activation of phospholipase (PLC), which hydrolyzes phosphatidylinositol 4, 5–diphosphate (PIP2) to inositol triphosphate (IP3) and diacylglycerol (DAG). The controlled entry of Ca^2+^ from the endoplasmic reticulum (ER) activates calmodulin (CaM) and heat shock signaling pathways [18]. The downstream signaling components, including protein kinases, e.g., calcium–dependent protein kinases (CDPKs) and mitogen activated protein kinases (MAPKs), activate heat shock transcription factors (HSFs) by protein phosphorylation. HSFs then activate heat shock proteins (HSPs) and other target proteins that are related to heat tolerance. HSF/HSP molecular modules are at the core of plant transcriptional regulatory networks upon HS.

It is well known that in the plant hormone–mediated regulation of heat tolerance in response to global climate change [19], the stress phytohormone ABA plays a key role [20]. Up–regulation of HSFA2c and HSPs by ABA contributes to the improved heat tolerance of various plant species [21]. The ABA–HSF/HSP signaling pathway was acted on in grafted cucumber plants [22], tall fescue and *Arabidopsis* [23], and developing grains in wheat [24]. ABA also exerted functions by reprogramming metabolism upon high–temperature stress, e.g., by mediating sugar metabolism in rice spikelets to prevent pollen abortion [25], through regulation of osmolytes and antioxidants to protect photosynthesis and growth in wheat plants [26], as well as to improve grain filling capacity in rice [27]. However, whether PYL/RCAR–mediated ABA signaling was involved in the heat stress response remains unclear so far. The functional characterization of the orthologous genes of PYL/RCAR ABA receptors in more crop plants, as well as exploration of their transcriptional regulation network, holds great promise for both sustainable agriculture and ecological systems, especially under global climate change [13].

Most *Lycium* (Goji) plants not only have medicinal and edible value but also have good resistance to adverse environments [28,29,30]. *Lycium* plays an important role in the utilization of marginal land due to its characteristics of drought, salt, and alkali resistance and barren resistance. However, global climate change–derived heat stress has greatly impeded the yield and quality of *L. barbarum*, one of the main popular cultivars around the world, in recent years [31]. Continuous high–temperature weather resulted in fruit falling, accelerated ripening, shortened picking intervals, advanced aestivation (a physiological phenomenon under high–temperature conditions in summer, that is, growth stops and leaves fall off and enter a dormancy state), and a delayed growth period of autumn twigs. At the same time, it also causes the occurrence of wolfberry diseases and pests [31]. In this case, exploring the molecular mechanism underlying the heat stress response becomes a vital issue for *Lycium* biology. The role of PYL/RCAR–mediated ABA signaling in the heat stress response of *Lycium* plants is still unclear, and 12 homologs of Arabidopsis PYL/RCAR ABA receptors were first identified from *L. barbarum* at a genome–wide scale in the present work. After a systematic bioinformatics analysis, a transcript profiling assay was conducted in response to ABA or heat stress. It is found that all of the genes were up–regulated by heat stress in a time–dependent manner. One of the genes, *LbPYL10*, which was transcriptionally activated both by ABA and heat stress, has been shown positively mediate the heat stress tolerance of *L. barbarum* seedlings by virus–induced gene silencing (VIGS) analysis. All these findings imply that the PYL/RCAR–mediated ABA signaling could play important roles in the heat stress response in *L. barbarum*, which provides important gene resources for the genetic improvement of the heat resistance of *L. barbarum* in a changing environment.

## 2. Results

### 2.1. Heat Stress–Induced ABA Accumulation in L. barbarum

In view of the role of ABA exerted in the heat tolerance of *L. barbarum*, we conducted a time–course ABA assay under 42 °C (Figure 1). It is shown that heat stress significantly enhanced ABA accumulation, though the response time was relatively slow. For the time course, there was not a significant change in the one–hour treatment, while the level was greatly raised after the 3 h treatment and then remain unchanged within the 24 h treatment time. This result highlights the potential role of ABA function in the heat stress tolerance of *L. barbarum* and promotes that we further explore its core signaling components.

### 2.2. Characterization of the LbPYL Gene Family

Twelve different genes encoding complete LbPYLs were identified both by BLAST searches in the *L. barbarum* reference genome using 12 Arabidopsis PYL amino acid sequences as queries (Table 1) and by conserved domain MIP PF00230 validation. The CDS, TMDs, pI, and MW of 12 LbPYLs were analyzed. As can be seen from the Table, most LbPYLs contain CDS between 534 and 747 base pairs in length. The predicted proteins range in length from 177 to 248 amino acids, with MWs of 20.229 to 27.636 kDa. The pI values vary between 5.34 and 8.65. These genes were unevenly distributed in all 12 chromosomes (Chr), except Chr02, Chr10, and Chr11, and detailed information is presented in Appendix A.

### 2.3. Phylogeny Analysis

To systematically classify *L. barbarum* PYL genes and uncover the evolutionary relationship with PYL genes from other plants, a phylogenetic tree was constructed with MEGA11 (11.0.13) using neighbor–joining (NJ) analysis (Figure 2). To study the evolutionary relationships between LbPYLs and PYLs from *Arabidopsis thaliana*, *Solanum lycopersicum*, *Populus trichocarpa*., and *Zea mays* L., the 12 LbPYLs with 14 AtPYLs, 14 SlPYLs, 14 PtPYLs, and 13 ZmPYLs were clustered into three groups, designated Group I, Group II, and Group III, on an unrooted phylogenetic tree. In this tree, with the exception of *LbPYL12*, *LbPYL1* and *LbPYL2* belonged to Group I, *LbPYL3*, *LbPYL4*, *LbPYL5*, *LbPYL10*, and *LbPYL11* belonged to Group II, and *LbPYL6*, *LbPYL7*, *LbPYL8*, and *LbPYL9* belonged to Group III (Figure 2). *LbPYL12* has high homology with *SlPYL13*, and previous reports have also indicated that *SLPYL13* (Sl02g076770) is independent of other subfamilies [32]. Therefore, we believe that LbPYL12 is also independent of other subfamilies.

### 2.4. Gene Structure, Conserved Motifs, and Domain Analysis

In order to fully understand the protein structure of LbPYLs, MEME (5.5.4) software was used to analyze the motif distribution of all 12 LbPYLs, and 15 conserved motifs were found (Figure 3B). Motifs 1, 2, and 3 were highly conserved across all LbPYLs. Whereas motifs 6 and 15 were only present in the Group I subfamily, motifs 4, 8, 9, and 10 were only present in the Group II subfamily, and motifs 5, 7, and 12 were only present in the Group III subfamily, except LbPYL9. Motif 11 was present in both LbPYL2 and LbPYL7, motif 13 was present in both LbPYL11 and LbPYL12, and motif 14 was present in both LbPYL9 and LbPYL5 (Figure 3A,B).

The intron–exon structure of LbPYLs was analyzed by TBtools (Figure 3D). The LbPYLs can be distinctly grouped into a clade with two introns and two other clades. Groups I and II have one intron or no intron, and all members of Group III have two introns. This result indicates that members of the subfamily have a highly similar genetic structure, which is consistent with their phylogenetic relationships (Figure 3C,D).

Similar to the PYR/PYL/RCAR proteins in *Arabidopsis*, all LbPYL proteins are composed of a highly similar helix–grip structure, characterized by two α–helixes (red) on either side of a seven–strand β–sheet (green) (Figure 4).

### 2.5. Analysis of Cis–Elements in the LbPYL Promoters

To better understand the transcriptional regulation and potential function of the LbPYL genes, we isolated sequences within 2000 bp upstream of the LbPYLs’ start codon and identified the *cis*–elements in these promoter sequences using the PlantCARE database. The major *cis*–elements included stress–related elements, such as anaerobic induction regulatory elements (AREs), dehydration stress elements (DRE cores), light responsive elements (G–boxes), MYBs, MYCs, and AAGAA motifs, as well as hormone–related elements, such as abscisic acid–responsive elements (ABREs) and ethylene–responsive elements (EREs). The promoter region of 12 LbPYL genes contains at least six *cis*–elements, and all LbPYL genes contain MYB and MYC elements. As can be seen from the statistical table, *LbPYL10* has the largest number of *cis*–acting elements, followed by *LbPYL11* (Figure 5).

### 2.6. Tissue–Specific Expression

To characterize the expression of the *LbPYL* gene family, we analyzed five different tissues of *L. barbarum* based on qRT–PCR (Figure 6). The different expression levels of LbPYL members in different tissues indicate that different tissues require different functions. Notably, most LbPYL genes are expressed at higher levels in the roots than in other organs. The genes *LbPYL3*, *LbPYL4*, *LbPYL5*, *LbPYL6*, *LbPYL*7, *LbPYL8*, *LbPYL9*, *LbPYL10*, and *LbPYL11* had maximal transcripts in roots rather than in other tissues. The *LbPYL1* and *LbPYL2* genes had the highest transcript levels in leaves, and *LbPYL12* had the highest transcript levels in fruits. No genes were preferentially expressed in flowers. Among the three subfamilies, subfamily II and subfamily III were preferentially expressed in roots, and subfamily I was expressed not only in roots but also in leaves (Figure 6). Taken together, these overlapping and preferential expression patterns of *LbPYLs* may allow *L. barbarum* to perform different ABA responses in specific tissues.

### 2.7. Transcriptional Response to Exogenous ABA

Under 10 μM ABA treatment, the transcript content decreased with the extension of the treatment time in 7 of the 12 members. Compared with the control, the expression levels of *LbPYL2*, *LbPYL4*, *LbPYL5*, and *LbPYL10* were up–regulated for a short time and then decreased (*LbPYL5* and *LbPYL10*) or remained unchanged (*LbPYL2* and *LbPYL4*) during the three–hour treatment (Figure 7). The expression of the *LbPYL3* gene continued to increase within three hours. It is clear that the up–regulated genes mainly belonged to the Group II subfamily, while those genes belonging to the Group I and Group III subfamilies were consistently down–regulated by exogenous ABA (Figure 7).

### 2.8. Transcriptional Response to Heat Stress

Within 24 h, the expression of the *LbPYL* gene first increased and then decreased in response to heat stress (Figure 8). The time dynamics of the responses are different for different subfamilies. While one–half of them (*LbPYL1*, *LbPYL2*, *LbPYL11*, *LbPYL4*, and *LbPYL9*) were up–regulated by 2~5–fold, the remaining genes (*LbPYL5*, *LbPYL3*, *LbPYL10*, *LbPYL7*, *LbPYL8*, *LbPYL6*, and *LbPYL12*) were up–regulated over 5–fold. In the latter, two genes, *LbPYL10* and *LbPYL12*, were up–regulated over 30–fold (variations in scale require special attention). In the time course, three genes (*LbPYL5*, *LbPYL3* and *LbPYL10*) obtained maximal expression at 1 h, while three genes (*LbPYL1*, *LbPYL2* and *LbPYL11*) obtained expression peaks at 3 h. The remaining genes (*LbPYL4*, *LbPYL7*, *LbPYL8*, *LbPYL6*, *LbPYL9*, and *LbPYL12*) had relatively slower response speeds (Figure 8). As you can see, members of subfamily II respond the fastest, followed by subfamily I, and subfamily III responds the slowest. These findings suggest that *LbPYL* members might play diverse roles in sensing the ABA signal for *L. barbarum* to adapt to heat stress.

### 2.9. Functional Validation of LbPYL10 in Heat Stress Tolerance

Considering that there is a significant transient up–regulation pattern for the *LbPYL10* gene in comparison to its homologues in *L. barbarum* under heat stress, we further validated its function in heat tolerance by VIGS technology (Figure 9). The 329 bp fragment of the *LbPYL10* gene was cloned into the TRV RNA2 vector (Figure 9A and Appendix A). After detecting *LbPYL10* expression in TRV–infected plants by RT–qPCR technology, the significantly *LbPYL10*–silenced lines were used for subsequent experiments (Figure 9B). Compared with the control, *LbPYL10* silencing significantly stimulated leaf disc bleaching and senescence following two hours of 42 °C treatment (Figure 9C).

In order to explore the mechanism underlying *LbPYL10* silencing–mediated leaf disc bleaching, the chlorophyll degradation–related genes were identified from *L. barbarum* at a genome–scale (Appendix A), and their expression pattens were analyzed in *LbPYL10*–silenced plants upon heat stress. It is shown that *LbPYL10* silencing significantly up–regulated pheophorbide a oxygenase (*LbPAO*) and stay–green proteins like *LbSGRL* expression (Figure 9D,E). *LbPAO* and *LbSGRL* were homologs of Arabidopsis NtPAO (EU294211.1) and AtSGRL (AT1G44000), respectively (Appendix A). Our results may highlight that LbPYL10 actively mediates the heat stress tolerance of *L. barbarum* by inhibiting chlorophyll degradation, thus delaying leaf senescence.

## 3. Discussion

The development of PYLs/RCARs, as an initial factor inducing ABA signaling, is considered to be a key event in the evolution of aquatic plants into terrestrial plants. No members of the PYL/RCAR family are present in the aquatic algae Chlamydomonas, while the number of PYL/RCAR family members increases from mosses and ferns to angiosperms. By far the largest numbers have been reported in *Musa acuminata* and *Brassica rapa* (24), and the lower plant *Marchantia polymorpha* had the least (four) [2]. A total of 14 and 23 members of PYL/RCAR ABA receptors have been identified in two Solanaceae families, tomato and tobacco, respectively [6,32]. In the present work, we found that 12 members of this family of proteins exist in one of the most well–known *Lycium* plants, *L. barbarum*, highlighting the high conservation of this family of genes. It is speculated that heat stress may indirectly regulate LbPYL gene expression by activating the upstream transcriptional factors, e.g., MYB, DRE, etc., due to the fact that no potential heat shock elements (HSEs) were identified in these genes. It should be noted that *L. barbarum* is only one of the eight species, and four varieties of *Lycium* plants exist in China [28,29,30]. Along with the pangenome published for more *Lycium* plants in the future, the number of PYL/RCAR ABA receptors may also vary within this plant species.

ABA takes part in the heat stress response and has also been reported in some plant species [19,20,21,23,33,34,35], in which the exogenous application of this hormone prior or parallel to heat stress renders plants more thermotolerant, although the signaling mechanism is not well documented. In this study, we found that most of the LbPYL transcripts showed rapid and sharp up–regulation under a high–temperature stress of 42 °C. Integrating the time course and amount of the transcriptional response, it is speculated that the genes *LbPYL3*, *LbPYL5*, and *LbPYL10*, which belong to subfamily II, may play a major role in the heat acclimation of *L. barbarum* seedlings. It is speculated that this subfamily’s members conservatively mediate plants’ acclimation to global climate changes, as the overexpression of its orthologs TaPYL4 and PePYL4 enhanced draught tolerance in wheat and populus, respectively [11,36]. Considering that the level of endogenous ABA increased following heat stress, and *LbPYL10* silencing significantly stimulated leaf disc bleaching and senescence, it may be concluded that LbPYL–mediated ABA signaling plays a vital role in the heat stress tolerance of *L. barbarum* plants.

As is widely acknowledged, photosynthesis is significantly impacted by aging or abiotic stress, and plants enhance their survival rate through nutrient recycling from damaged organelles [37,38]. It has been reported that high temperature can disrupt the stability of the light–harvesting chlorophyll–protein complex and result in the deactivation of photosynthetic enzymes. Additionally, under high–temperature stress, there was a significant increase in the expression of genes associated with chlorophyll degradation (*PPH*, *ACD2*, *NYE*, and *HCAR*) in *Arabidopsis thaliana* [39,40]. The crucial role of *AtPAO* in chlorophyll degradation during leaf aging has previously been identified in *Arabidopsis thaliana*; it is believed that Pheide a degradation ultimately leads to leaf bleaching [41]. Furthermore, studies have demonstrated that *Arabidopsis* plants overexpressing *AtSGRL* exhibit earlier leaf bleaching under abiotic stress conditions while leaves of *atsgrl*–*1* mutants remain green for an extended period [42]. In our current study on *L. barbarum*, we examined the relative expression levels of *LbPAO* and *LbSGRL*. We observed higher expression levels in *LbPYL10*–silenced plants under high–temperature stress. Silencing *LbPYL10* strengthened the chlorophyll degradation pathway and accelerated leaf bleaching after an equivalent duration of high–temperature stress. This promoted the process of leaf senescence. In conclusion, *LbPYL10* positively regulates heat stress tolerance in *L. barbarum* by suppressing the chlorophyll degradation pathway and maintaining stability within the chlorophyll–protein complex.

In the future, the further characterization of the downstream signaling molecules of LbPP2Cs and LbSnRK2s, followed by exploration of the LbPYL–LbPP2C–LbSnRK2 combinations, will greatly advance our understanding of LbPYL–mediated ABA signaling in *Lycium* plants, as complicated combinations have been shown in *Arabidopsis*, maize, and tomato [9,43,44,45,46]. Moreover, establishing an efficient CRISPR/Cas9 gene editing system for *L. barbarum* will greatly accelerate the genetic improvement of *L. barbarum* for climate adaptation, such as *LbPYL10*. Finally, it should be noted that the reproductive growth stage is usually the most sensitive stage to heat stress; therefore, exploring LbPYL–mediated ABA signaling in this stage would provide more valuable genetic information.

## 4. Materials and Methods

### 4.1. Plant Materials

The different tissues, including young roots, stems, leaves, and flowers, as well as the ripening fruits of *Lycium* (L.) *barbarum*, were collected from three of the 5–year–old trees at the Wolfberry (*Lycium*) Germplasm Repository of Ningxia, Academy of Agriculture and Forestry. Our field studies were conducted in accordance with local legislation and appropriate permissions.

### 4.2. Heat Stress Assay

Two–month–old uniform clonal seedlings of *L. barbarum*, which were grown in pots in a greenhouse, were transported to a climate chamber. After acclimation to the artificial environment (25 °C, 16 h day/20 °C, 8 h night) for one week, the seedlings were divided into two parts. Whereas one part was still left in the same climate chamber as the control, the other part was transported to a 42 °C growth chamber for heat stress. Other environmental factors, such as light intensity (150 μmol∙m^−2^∙s^−1^) and humidity (65%), remained consistent between the two climate chambers. The seedlings were subjected to heat stress for 0, 1, 3, 6, 12, and 24 h. The leaves were sampled at each time point and immediately frozen by liquid nitrogen for RNA extraction. Three biological replicates were set for each treatment.

The function of LbPYL10 in heat stress tolerance was validated by leaf disc assay, as described by Lee [47] with some modifications. Discs (5 mm in diameter) were punched from the second to fifth fully expanded leaves of 25–day–old soil–grown plants and then floated on 2 mL of 10 mM MES–KOH buffer, pH 6.8, on 12– or 24–well microplates. Plates were incubated at 42 °C for 2 h. After heat stress treatment, leaf discs were returned to 25 °C under 16 h of light/8 h of dark and photographed 5 d later.

### 4.3. ABA Treatment

One–month–old uniform seed seedlings of *L. barbarum*, which were cultured in 1/2 Hoagland nutrient solution in a climate chamber, were subjected to 10 μM ABA for 0 (Mock), 0.5, 1, and 3 h, respectively. The climate chamber conditions were as follows: light intensity (150 μmol·m^−2^·s^−1^), humidity (65%), 25 °C, 16 h day/20 °C, and 8 h night. The ABA storage solution (dissolved in methanol) was diluted in 1/2 Hoagland nutrient solution. The leaves were sampled at each time point and immediately frozen by liquid nitrogen for RNA extraction. Three biological replicates were set for each treatment.

### 4.4. ABA Content Determination

The ABA content was determined as described by Guo [48]. Fifty milligrams (fresh weight) of the sample powder was extracted with 1 mL methanol/water/formic acid (15:4:1, *v*/*v*/*v*). The combined extracts were evaporated to dryness under a nitrogen gas stream, followed by being resuspended in 100 μL 80% methanol (*v*/*v*), as well as filtered through a 0.22 μm filter for further LC–MS analysis. The sample extracts were analyzed using a UPLC–ESI–MS/MS system (UPLC ExionLC™ AD https://sciex.com.cn/, accessed on 27 August 2023; MS Applied Biosystems 6500 Triple Quadrupole, https://sciex.com.cn/, accessed on 27 August 2023).

### 4.5. Virus–Induced LbPYL10 Gene Silencing

The CDS fragment of the *LbPYL10* gene carrying the adaptor sequence was obtained by a pair of amplification primers (Forward: 5′–AAGAGCTGCCACATCATCGG; Reverse: 5′–GCGATCTGGGAAAGGGATTG). The pTRV2 vector was digested with *Eco*R I and *Bam*H I to not only linearize the vector but also expose the adaptor sequence. The PCR product was connected to the sticky end of the vector using a DNA OK Clon DNA Ligation Kit (Accurate Biotechnology, Changsha, China) and then transformed into *E. coli* to screen for pTRV2–derivative colonies with the silenced target gene (pTRV2–*LbPYL10*). After pTRV1 and pTRV2 or pTRV2–*LbPYL10* were rapidly frozen in liquid nitrogen, they were transfected with the Agrobacterium tumefaciens strain GV3101 by heat shock.

The infection method was performed as previously described [49]. Agrobacterium strains of pTRV1, pTRV2, and pTRV2–*LbPYL10* were collected and resuspended in infiltration buffer (10 mM MgCl_2_, 10 mM MES, and 200 μM acetosyringone) at an optical density of 0.3 at 600 nm, and then suspensions were maintained at room temperature for at least 3 h. Finally, pTRV2 or pTRV2–*LbPYL10* was mixed at a 1:1 ratio with pTRV1 before injection. One–month–old seedlings were prepared and injected using the pressure permeation method. After 2 min of injection, the surface was cleaned with ddH_2_O. After 2–3 weeks, the silencing efficiency in pTRV2–*LbPYL10* plants was analyzed using qRT–PCR.

### 4.6. Identification and Chromosomal Location of LbPYL Genes

To identify the putative PYL proteins in *L. barbarum* (LbPYLs), the 14 AtPYR/PYLs in the *A. thaliana* genome were downloaded from the TAIR database (https://www.arabidopsis.org/, accessed on 5 February 2023) and were then used as queries to BLAST search the *L. barbarum* genome (https://www.ncbi.nlm.nih.gov/genome/81199?genome_assembly_id=1656998, accessed on 23 December 2023) with an E–value of e^−10^. Then, preliminary amino acid sequences that may function as LbPYLs were obtained according to the homology of AtPYR/PYLs. On this basis, conservative PYL domains obtained from the PFAM database (http://pfam-legacy.xfam.org/, accessed on 1 May 2023) were blast searched against these candidate sequences. Finally, amino acid sequences without conserved PYL domains and redundant sequences were manually removed.

The values of the molecular weights (MWs) and isoelectric points (pIs) of the LbPYL proteins were analyzed with ProtParam (http://web.expasy.org/protparam/, accessed on 20 January 2023).

The chromosomal locations of LbPYL genes were displayed with TBtools (v2.057) software [50] based on the genome annotation files downloaded from the Genome Database for *L. barbarum*, in which the chromosome numbers and positions of each sequence in the genome are indicated.

### 4.7. Phylogenetic Tree Analysis of LbPYLs in L. barbarum

Sequences of PYLs from *A. thaliana*, S. *lycopersicum*, *Populus* L., and *Zea mays* L. were acquired from Ensemble Plants (http://plants.ensemble.org/index.html, accessed on 5 February 2023). The maximum likelihood tree (ML tree) was constructed by the neighbor–joining (NJ) method with MEGA11 (11.0.13) software derived from a ClustalW2 alignment of related amino acid sequences (bootstrap replicates = 1000). The ML tree was beautified by the ChiPlot website (https://www.chiplot.online/#, accessed on 7 February 2023) [51].

### 4.8. Analysis of Gene Exon–Intron Structures and Protein Conserved Motifs

TBtools (v2.057) software was used to draw the structure map. The conserved motif analysis was generated by the classic mode of MEME (Multiple Em for Motif Elicitation, https://meme-suite.org/meme/, accessed on 2 May 2023), where the number of motifs was set to 15 and other settings were consistent with the default parameters.

### 4.9. Analysis of Cis–Elements in the LbPYL Promoters

The promoter regions 2 kb upstream of the corresponding gene were analyzed, and the *cis*–elements were predicted by the PlantCARE database (https://bioinformatics.psb.ugent.be/webtools/plantcare/html/, accessed on 8 March 2023). Subsequently, according to the results of the PlantCARE calculation, we recorded the numbers of each *cis*–element in these sequences and summarized this information in a figure for subsequent analysis.

### 4.10. Quantitative Real–Time PCR (qRT–PCR) Assay

Total RNA was isolated from different tissues using TRIzol reagent (Invitrogen, Carlsbad, CA, USA), and its quality was examined by NanoDrop (Thermo Scientific, Waltham, MA, USA). First–strand cDNAs were synthesized from 1 μg of RNA using the FastKing RT Kit (TIANGEN, Beijing, China). qRT–PCR was conducted using 0.1 μg of cDNA on a CFX96 Touch (BIO–RAD) with SuperReal PreMix Color (SYBR Green) and with an initial denaturing step at 95 °C for 15 min, followed by 40 cycles of 95 °C for 10 s, 60 °C for 30 s, and 72 °C for 32 s. The PCR primers used for qRT–PCR are listed in Appendix A.

### 4.11. Statistical Analysis

Statistical analyses were performed using SPSS version 19.0. Parameter differences were determined using one–way ANOVA (Duncan’s test) with appropriate post hoc analysis. In some cases, significant differences in mean values were determined by Student’s *t*–test and different letters indicate significant differences at *p* < 0.05. The column and line figures were drawn by Origin 2023b.

## 5. Conclusions

This study presented a comprehensive genome–wide analysis of the PYR1/PYL/RCAR ABA receptor gene family in *L. barbarum*. The *LbPYL* gene expression profiling analyzed by qRT–PCR revealed high expression in roots versus other tissues and differential sensitivity to exogenous ABA or heat stress. It is speculated that the genes *LbPYL3*, *LbPYL5*, and *LbPYL10*, which belong to subfamily II, may play a major role in the heat stress acclimation of *L. barbarum* seedlings. *LbPYL10*, which was instantaneously up–regulated by heat stress or ABA treatment, positively regulated heat stress tolerance by alleviating chlorophyll degradation, thus maintaining chlorophyll stability. Taken together, our results highlight a vital role of LbPYL ABA receptor–mediated ABA signaling in *L. barbarum’s* response to heat stress.

## Figures and Tables

**Figure 1 plants-13-00887-f001:**
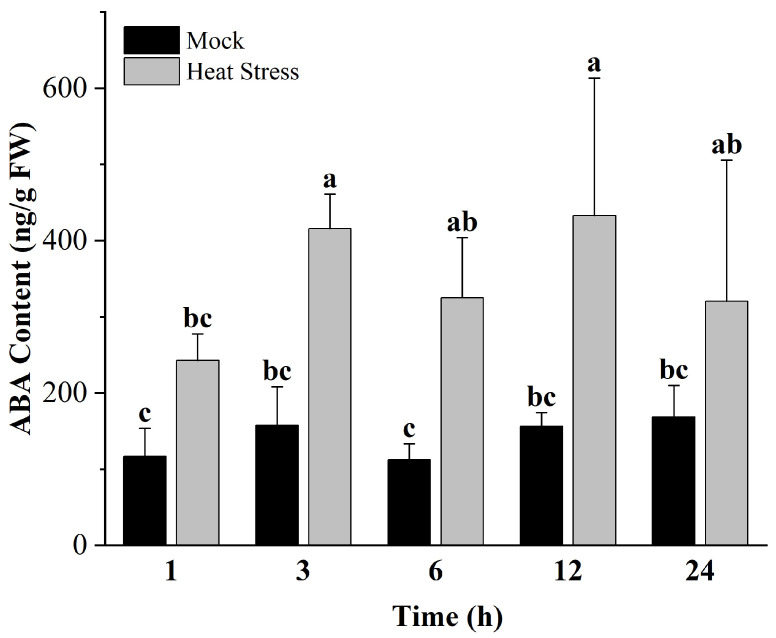
ABA level response to heat stress. The ABA content was measured in the leaves of 2–month–old *L. barbarum* Ningqi No.7, which was subjected to an artificial environment and 42 °C for 1, 3, 6, 12, or 24 h. Different letters indicate significant differences at *p* < 0.05 (Duncan’s test).

**Figure 2 plants-13-00887-f002:**
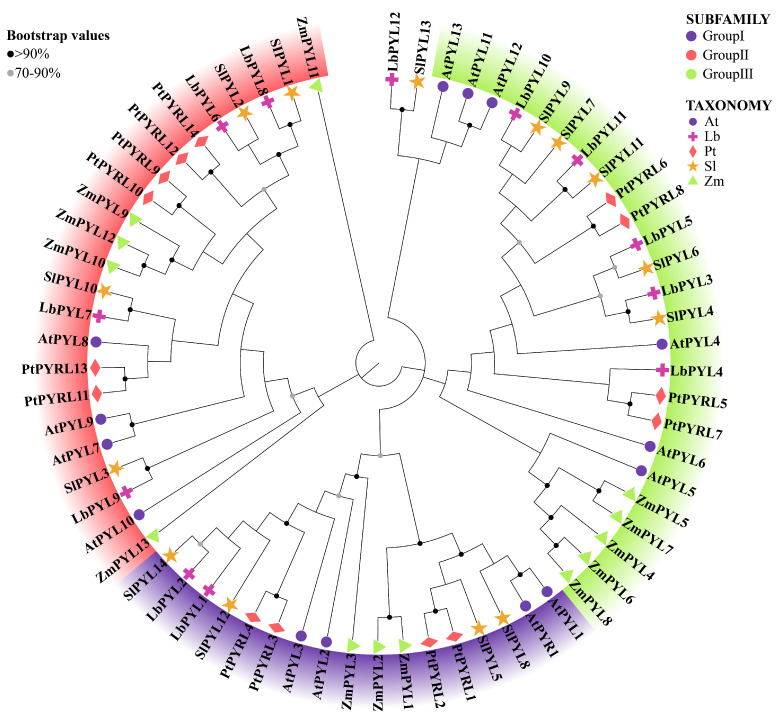
Phylogenetic analysis of the 12 *LbPYLs* with Arabidopsis and tomato homologs. Deduced amino acid sequences were aligned using ClustalX, and the phylogenetic tree was constructed using the bootstrap maximum likelihood tree (1000 replicates) method and MEGA11 (11.0.13) software. The full–length PYL protein sequences, including 12 members from *L. barbarum* (Lb), 14 members from *A. thaliana* (At), 14 members from S. *lycopersicum* (Sl), 14 members from *Populus trichocarpa* (Pt), and 13 members from *Zea mays* L. (Zm), were classified into the I, II and III subfamilies. The branches of different classes have altered colors, and each represents a different subfamily. The sequence file information is presented in Appendix A.

**Figure 3 plants-13-00887-f003:**
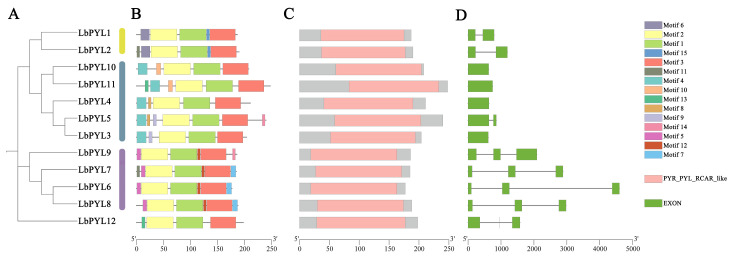
Phylogenic relationship, protein structure conserved domains, and conserved motif analyses. (**A**) A phylogenetic tree of LbPYL proteins was constructed with MEGA11 (11.0.13) software. (**B**) Conserved motif distribution of LbPYL proteins. The conserved motifs identified with MEME are displayed with boxes in different colors. A total of 15 motifs were identified and the motifs’ sequences are described in Appendix A. The scale at the bottom shows the length of the protein. (**C**) Predicted conserved structural domains of LbPYL proteins. Gray lines represent the length of each protein sequence, and conserved domains are indicated by colored boxes. (**D**) Exon–intron structures of the LbPYL genes. Gene structures were analyzed with Tbtools (v2.057) software. Exons and introns are indicated with green boxes and gray lines, respectively. The lengths of the exons and introns (bp) are indicated on the *x*–axis. The combined figure was illustrated with Tbtools (v2.057) software.

**Figure 4 plants-13-00887-f004:**
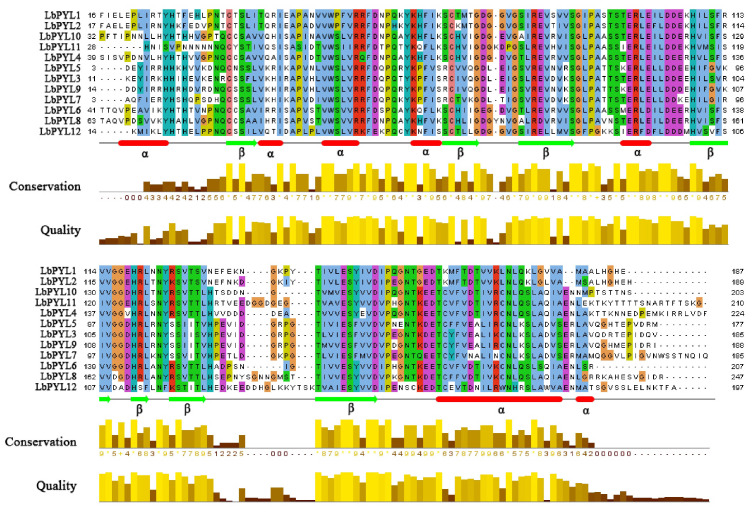
Protein sequence alignment of LbPYLs. Conserved transmembrane domains (TM1–6) and amino acids at NPA domains, ar/R selectivity filters, and Froger’s residues identified in LbPYLs.

**Figure 5 plants-13-00887-f005:**
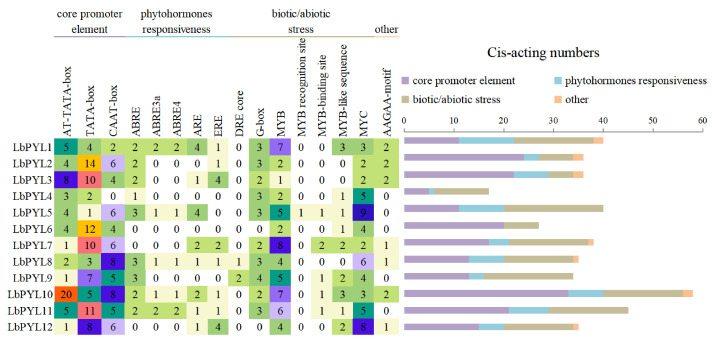
The *cis*–acting element on the putative promoter of the LbPYL genes. The identified *cis*–acting elements were mainly divided into three major categories: phytohormone responsiveness, biotic/abiotic stress, and others. The *cis*–acting elements sequences are described in Appendix A.

**Figure 6 plants-13-00887-f006:**
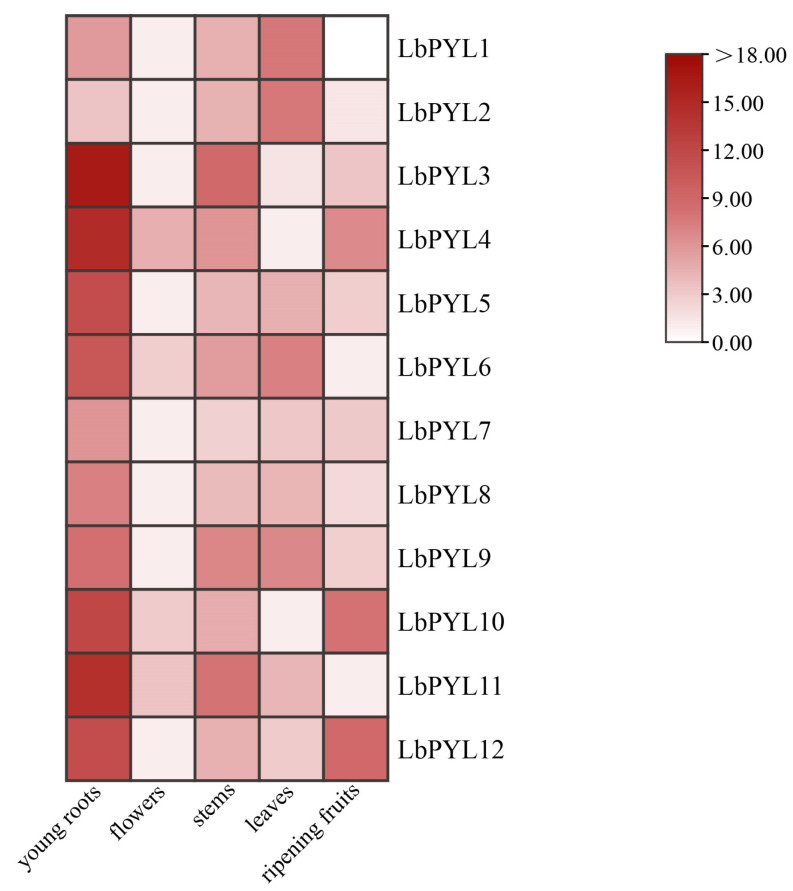
Tissue–specific expression of LbPYL genes. Tissue–specific expression of LbPYLs was determined by quantitative real–time PCR (qRT–PCR) in leaves, young roots, stems, ripening fruits, and flowers with gene–specific primers. qRT–PCR was performed in triplicate, and the fold change was analyzed via the 2^−ΔΔCT^ method using the LbACTIN1 gene as an internal control. Values are the means of three independent experiments.

**Figure 7 plants-13-00887-f007:**
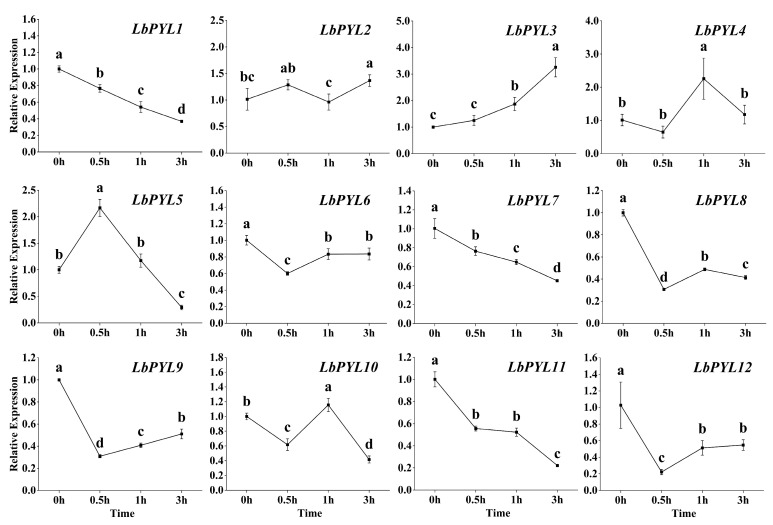
Expression profiling response to ABA. The expression levels of LbPYLs were measured in the leaves of one–month–old *L. barbarum* subjected to 10 μM ABA for 0, 0.5, 1, or 3 h. The fold change in the relative expression level was analyzed via the 2^−ΔΔCT^ method using the LbACTIN1 gene as an internal control. Values are the means of three independent experiments. Different letters indicate significant differences at *p* < 0.05 (Duncan’s test).

**Figure 8 plants-13-00887-f008:**
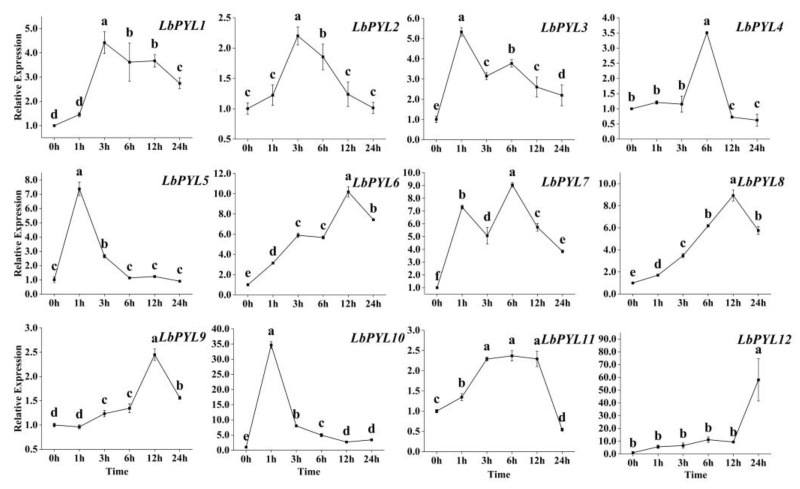
Expression profiling response to heat stress. The expression levels of LbPYLs were measured in the leaves of one–month–old *L. barbarum* subjected to 42 °C for 0, 1, 3, 12, or 24 h. The fold change in the relative expression level was analyzed via the 2^−ΔΔCT^ method using the LbACTIN1 gene as an internal control. Values are the means of three independent experiments. Different letters indicate significant differences at *p* < 0.05 (Duncan’s test).

**Figure 9 plants-13-00887-f009:**
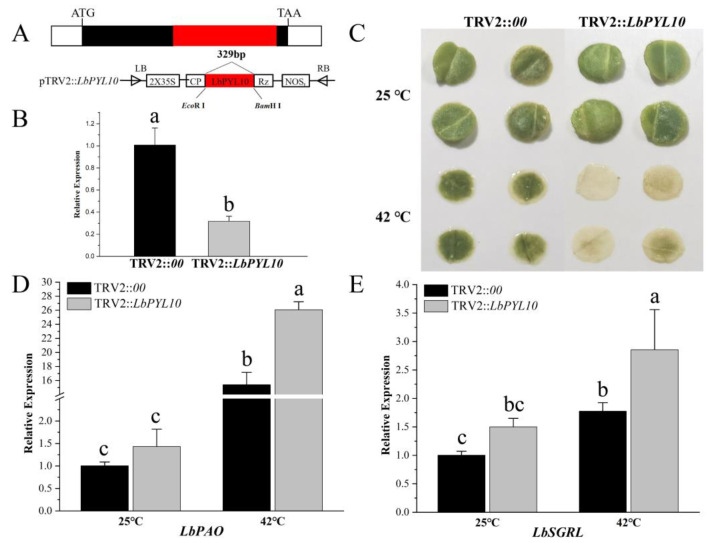
Validation in *LbPYL10* function by VIGS technology. (**A**) *LbPYL10* gene structure in *L. barbarum* and pTRV2– *LbPYL10* vector map used in this study. White boxes represent 5′– and 3′–UTRs, black boxes represent CDS, and red boxes represent target silent fragment. (**B**) Expression of *LbPYL10* in *LbPYL10*–silenced (TRV2::*LbPYL10*) and control (TRV2::*00*) plants, respectively. (**C**) Leaf disc phenotype difference between the *LbPYL10*–silenced seedlings and the empty vector–infected seedlings after heat stress. (**D**,**E**) Effect of *LbPYL10* silencing on pheophorbide a oxygenase (*LbPAO*, XM_060313688.1) and STAY–GREEN–like (*LbSGRL*, XM_060342386.1) after heat stress. Values are the means of three independent experiments. Different letters indicate significant differences at *p* < 0.05 (Duncan’s test).

**Table 1 plants-13-00887-t001:** Detailed information on 12 PYL genes of *L. barbarum* and their encoded proteins.

Gene Name	Version	Gene Length (bp)	CDS Length (bp)	Deduced Protein		
				Size (aa)	MW (kDa)	PI
LbPYL1	XM_060355288.1	1356	564	187	21.045	6.16
LbPYL2	XM_060315403.1	1640	573	190	21.305	5.74
LbPYL3	XM_060316745.1	970	615	204	22.646	6.23
LbPYL4	XM_060357010.1	992	636	211	24.401	6.29
LbPYL5	XM_060322965.1	1315	723	240	26.651	5.52
LbPYL6	XM_060317057.1	5513	534	177	20.229	6.38
LbPYL7	XM_060347829.1	3489	558	185	21.078	5.75
LbPYL8	XM_060350890.1	3670	567	188	21.459	6.30
LbPYL9	XM_060362463.1	2092	561	186	20.762	5.96
LbPYL10	XM_060346502.1	1178	627	208	22.874	8.18
LbPYL11	XM_060320401.1	1103	747	248	27.636	8.65
LbPYL12	XM_060337877.1	2052	597	198	22.276	5.34

## Data Availability

Data are contained within the article.

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
