# Peer review of "Genome-Wide Identification of PYL/RCAR ABA Receptors and Functional Analysis of LbPYL10 in Heat Tolerance in Goji (Lycium barbarum)"

_plants, 2024, doi:10.3390/plants13060887_

Round 1
Reviewer 1 Report
Comments and Suggestions for Authors
Li et al. have identified the family of PYL/RCAR ABA receptors in Lycium barbarum, characterized their expression in response to ABA and heat stress, and used VIGS to knock down expression of 1 family member. They found differential expression of the various family members: some were induced, while others were repressed, by ABA. Although all were induced by heat stress, the kinetics and magnitude varied among family members. They knocked down the family member with the fastest strong response to heat stress, LbPYL10, and found that this made the leaves hypersensitive to heat stress, resulting in chlorophyll degradation and consequent bleaching of leaf tissue.
The manuscript would benefit from several revisions:
Lines 113-120 are very redundant. It’s not necessary to repeat all the size/mass details.
Lines 123-127: describe genes as “unevenly distributed” and then “evenly distributed”. Essential info can be stated more clearly.
Figure 4 identifies 15 motifs, but none are ever actually described in either the main document or the supplementary figures. In fact Supp Fig 1 (“conserved motifs”) just shows more undefined motifs for PAO and SGRL.
Line 174: refers to “others” in the PYL genes, but not clear what “others” are
Fig.6 shows many MYB elements in all LbPYL putative promoters, and 3 subclasses (?) of MYBs almost absent from those same sequences. Although probably defined in the PlantCARE database, it would be helpful to identify the relevant sequences for this cis elements, possibly in the supplemental data
Line 202: says LbPYL expression was analyzed in RNAseq datasets (Fig. 7), but Fig. 7 shows results of qRT-PCR, not RNAseq.
Line 227: what does “conformably” mean?
Lines 243-244: overgeneralizes subfamily response profiles; not all members of a given subfamily show the described response. These subfamily patterns, for both ABA and heat stress, are also emphasized in the abstract. It would be good to discuss whether this is a conserved response pattern for these family members, compared to their likely orthologs in other species.
Comments on the Quality of English Language
Would benefit from editing grammar throughout
Reviewer 2 Report
Comments and Suggestions for Authors
General Comments:
The manuscript presents a comprehensive study on the identification and functional analysis of PYL/RCAR ABA receptors in Lycium barbarum (goji), with a focus on their role in heat stress tolerance. The research is well-structured, addressing a significant gap in the understanding of ABA signaling pathways in goji plants, which are of economic, medicinal, and ecological importance. The identification of 12 homologs and the detailed analysis of LbPYL10's role in heat tolerance through VIGS technology provide valuable insights. However, there are several areas where the study could be improved to strengthen the findings and conclusions.
Specific Comments:
Introduction:
The introduction provides a good background on the role of ABA and PYL/RCAR receptors in plant stress responses. However, it would benefit from a more detailed discussion on the specific challenges that heat stress poses to plants (from heat perception to heat defenses, such as https://doi.org/10.1016/j.tibs.2022.05.004, https://doi.org/10.1111/pce.13979; https://doi.org/10.1016/j.tplants.2022.03.006) and to Lycium barbarum and the potential economic and ecological impacts of these challenges. This context would help to underscore the significance of the study.
Methods: explain why 42°C as a heat treatment.
Specific Comments:
Figure 1 (ABA Content):
- The figure supports the claim that heat stress induces ABA accumulation in L. barbarum. The error bars indicate some variability, and the different letters denote statistical differences, which is good practice. However, the authors should clarify the statistical method used for these analyses in the figure legend, as the inclusion of such information is critical for assessing the figure and the results. Same with all the others figures using statistics.
Figure 2 (Physical distribution of LbPYLs on chromosomes):
- The distribution of LbPYL genes across chromosomes is well depicted. The figure is clear and supports the claim of uneven distribution, which could imply functional divergence. But the authors should explain a bit more the relevance of this information. If it is not so important, this figure can move in supplemental figures.
Figure 3 (Phylogeny analysis):
- The phylogenetic tree presented is comprehensive, showing the evolutionary relationship between LbPYLs and PYLs from other species. The use of color-coding for subfamilies and different species is helpful. The bootstrap values are included, which is commendable.
Figure 4 (Gene structure, motifs, and domain analysis):
- This figure illustrates the motif composition and gene structure of the LbPYLs, showing clear differentiation between subfamilies. Additionally, this figure could be enhanced by alphafold structures of LbPYLs (in supplemental figures)
Figure 6 (Analysis of cis-elements in LbPYL promoters):
- This figure offers insights into the potential regulatory elements in the promoters of LbPYL genes, which is crucial for understanding gene regulation. The categorization of cis-elements and the visual representation are clear. However, HSE (heat shock elements) are missing despite they are very important cis-elements in regard to heat stress studies. I highly recommend to check for HSE and to discuss it in the Discussion part.
Figure 7 (Tissue-specific expression):
- This heatmap provides a snapshot of the differential expression of LbPYL genes in various tissues. I would recommend to replace the letters on x axis by name of the tissues.
Figure 8 (Transcriptional response to exogenous ABA):
- The graph shows the response of the LbPYL genes to ABA treatment over time. The data supports the differential response of the gene family to ABA, which is integral to the manuscript's claims. The inclusion of error bars is good for illustrating variability. However, the scale of the y-axis varies between graphs, which could mislead readers regarding the relative expression levels. Consistency in scale would be preferable. Regarding statistical test used, it should be put in down in the figure legend.
Figure 9: it is missing some control with very well know marker of heat stress response such as HSP70, HSP90, HSP20 or even HSFA2.
Figure 10 (Functional validation of LbPYL10):
- This figure is critical as it shows the functional validation of LbPYL10's role in heat stress. The authors have provided a visual representation of the VIGS experiment, gene expression analysis, and the physiological response to heat stress. The data presented supports the conclusions drawn regarding LbPYL10's function in heat tolerance. However, it's not clear whether the selected lines have been checked for VIGS effectiveness every time. In fact, although this approach has many advantages, it lacks consistency between generations (sometimes we get 80% inhibition, sometimes 50% and even between plants of the same generation it's random). Could the authors elaborate on this aspect?
Other remarks:
Functional Validation of LbPYL10: The authors have shown that silencing of LbPYL10 leads to increased leaf disc bleaching and senescence under heat stress. However, it is not clear whether this phenotype is specifically due to the loss of LbPYL10 function or a general response to gene silencing. It would be helpful to include a control where a non-related gene is silenced and check for similar phenotypes.
Role of Other LbPYLs: While the focus of the paper is on LbPYL10, it would be interesting to know the roles of other LbPYLs in heat stress response. Are there any other LbPYLs that show a similar or contrasting response to heat stress? This information can provide a more comprehensive view of the role of PYL/RCAR ABA receptors in heat stress response in L. barbarum.
Mechanism of LbPYL10 Action: The authors have shown that LbPYL10 silencing leads to upregulation of pheophorbide a oxygenase (LbPAO) and stay-green proteins like (LbSGRL). However, the mechanism through which LbPYL10 regulates these genes is not clear. Are these genes direct targets of LbPYL10 or is their regulation indirect? Further experiments, such as ChIP-qPCR, could help elucidate the mechanism of LbPYL10 action.
Comments on the Quality of English Language
English is fine, just re-check carefully.
Round 2
Reviewer 2 Report
Comments and Suggestions for Authors
Thanks for the replies.